# 2SSP: A Two-Stage Framework for Structured Pruning of LLMs

**Fabrizio Sandri**[*]                                            *sandri.fabr@gmail.com*
*University of Trento, Italy*

**Elia Cunegatti**[*]                                             *elia.cunegatti@unitn.it*
*University of Trento, Italy*

**Giovanni Iacca**                                               *giovanni.iacca@unitn.it*
*University of Trento, Italy*

**Reviewed on OpenReview:** *https://openreview.net/forum?id=Qd7LzJBg21*

## Abstract

We propose a novel Two-Stage framework for Structured Pruning (2SSP) for pruning Large Language Models (LLMs), which combines two different strategies of pruning, namely Width and Depth Pruning. The first stage (Width Pruning) removes entire neurons, hence their corresponding rows and columns, aiming to preserve the connectivity among the pruned structures in the intermediate state of the Feed-Forward Networks in each Transformer block. This is done based on an importance score measuring the impact of each neuron on the output magnitude. The second stage (Depth Pruning), instead, removes entire Attention submodules. This is done by applying an iterative process that removes the Attention with the minimum impact on a given metric of interest (in our case, perplexity). We also propose a novel mechanism to balance the sparsity rate of the two stages w.r.t. to the desired global sparsity. We test 2SSP on four LLM families and three sparsity rates (25%, 37.5%, and 50%), measuring the resulting perplexity over three language modeling datasets as well as the performance over six downstream tasks. Our method consistently outperforms five state-of-the-art competitors over three language modeling and six downstream tasks, with an up to two-order-of-magnitude gain in terms of pruning time. The code is available at https://github.com/FabrizioSandri/2SSP.

## 1 Introduction

The sheer size of the recent, billion-scale Large Language Models (LLMs) is one of the main reasons for their successful performance. However, it comes at the cost of computational budget in terms of required GPUs as well as time for pre-training and inference, which in turn has serious economic and environmental impacts. Therefore, studying approaches to reduce the computational burden of such models while minimizing their performance degradation has become a pressing matter.

One of the main approaches to address this issue is through Network Pruning (Frantar & Alistarh, 2023; Ma et al., 2023), which mainly focuses on reducing the size of pre-trained LLMs as well as their inference time. Among the several pruning methods available in the literature, reliable inference speed-ups (Kurtic et al., 2023; Ashkboos et al., 2024) have been achieved mainly through *structured* pruning, i.e., approaches that remove entire portions of the model. Different strategies to select which portions of the network to remove have been proposed, see Figure 1, identifying *Width pruning*, which removes rows/columns and/or single layers, *Depth Pruning (Blocks)*, which removes entire Transformer Blocks, and *Depth Pruning (Submodules)*, which removes entire submodules (i.e., Attention submodules—in the following, referred to as "Attentions",

---

[*]These authors contributed equally to this work.

for brevity—and/or Feed-Forward Networks (FFNs)) from the Transformer Blocks. *width* pruning has the main advantage of having a lower granularity level in the removal search space, which leads to a more refined identification of unimportant components of the model. On the other hand, the advantage of *depth* pruning lies in the lower computation time required to obtain the sparse structure as well as the larger inference speed-up that comes from the removal of entire blocks/submodules.

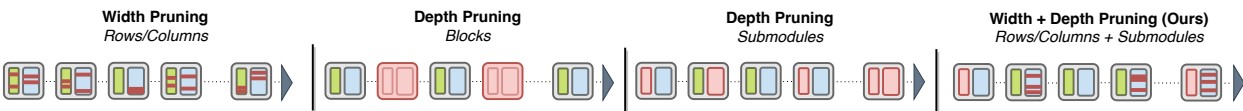

Figure 1: The four categories of structured pruning discussed. Each squared box indicates a Transformer block, with the Attention on the left and the FFN on the right. The red color indicates the elements pruned in each category, ranging from rows/columns to entire blocks or submodules therein. Our method 2SSP is the first one to combine width and depth (submodule) pruning.

**Contributions** So far, the aforementioned categories of structured pruning have been investigated independently of one another, and their combination is currently a relatively unexplored research direction. In this paper, we propose a Two-Stage Framework for Structured Pruning (2SSP), a new structured pruning approach that, to our knowledge, is the first to combine *width* and *depth* pruning, hence exploiting the advantages of both approaches. The proposed 2SSP works in two stages. The first stage works at a lower granularity level, i.e., it uses *width* pruning to remove neurons from the intermediate state of FFNs based on the output's magnitude. This is done while preserving the network connectivity, which is a critical measure for reducing performance degradation in sparse structures (Vysogorets & Kempe, 2023; Cunegatti et al., 2024; Iurada et al., 2024). Moreover, the first stage is applied only to the FFN parts of the model, for which pruning is known to be more difficult than Attention blocks (Siddiqui et al., 2024). The second stage complements the effect of the first one by iteratively applying *depth pruning* on Attentions, based on the minimization of a given performance metric (in our case, perplexity).

We tested our proposed method over four different LLM families, for three different sparsity rates (25%, 37,5%, and 50%) over three language modeling datasets, and six downstream tasks, showing that it consistently outperforms the five most recent state-of-the-art baselines while also having the best performance vs. pruning runtime trade-off. We also included experiments on higher sparsity levels, up to 70%, to further validate the effectiveness of our proposed method w.r.t. the baselines.

## 2 Related Work

In this section, we discuss the different techniques proposed in the literature to remove parameters from pre-trained LLMs (i.e., for pruning after training). Methods for removing parameters from a model, while minimizing its performance degradations, can be roughly categorized into *unstructured* and *structured* pruning. The main difference between these two approaches is that for a given sparsity and task, unstructured pruning allows for achieving better performance for the task, but the real speed-up is mainly theoretical (since the parameters are only masked, rather than practically removed); whereas, using structured pruning (the focus of this paper), the speed-up at inference/training time is substantial (because of actual structure removals), but the performance degradation on the task may be higher.

### 2.1 Unstructured Pruning

Unstructured pruning aims to reduce the model's parameters by identifying critical weights (inside a weight matrix $\mathbf{W}$), in any position, e.g., based on weight magnitude (Jaiswal et al., 2024), Hessian matrix (Frantar & Alistarh, 2023), or activations (Sun et al., 2024; Zhang et al., 2024a; Farina et al., 2024). On top of these methods, several techniques have been proposed either to select the best block-wise sparsity allocation, e.g., based on outliers (Yin et al., 2024), activation alignment (Cunegatti et al., 2025), and optimal allocation search (Li et al., 2024), or to further minimize the reconstruction error (Zhang et al., 2024b; Xu et al., 2024).

## 2.2 Structured Pruning

Structured pruning removes entire parameter groups or structures from the LLM. Different categories of structured pruning algorithms have been proposed, which can be roughly categorized w.r.t. the granularity of the structures they remove. The earliest approach is named **Width Pruning**, which encompasses methods that remove specific portions of weight matrices (such as rows and/or columns), or layers inside each Transformer block (such as a single Linear layer). The pruning decision can be based on gradient information (Ma et al., 2023; Fang et al., 2024), computational invariance (Ashkboos et al., 2024), perturbative pruning (Dery et al., 2024), hardware-aware inference vs. performance trade-off (Kurtic et al., 2023), Fisher information (van der Ouderaa et al., 2024), or knowledge-distillation (Muralidharan et al., 2024).

Recently, a new paradigm of pruning, namely **Depth Pruning**, has emerged, shifting the pruning focus from weight matrices and layers to either entire Transformer blocks or entire Transformer submodules (Attention and/or FFN). The main idea behind this category of pruning algorithms is to assign an importance score to each block/submodule and then prune it w.r.t. to its given score. These scores can be given either based on the change in hidden representation between blocks (similarity-based) or by computing each block's importance w.r.t. the model performance on a given task (performance-based).

Among the **Depth Pruning (Blocks)**, most of the existing methods are similarity-based (Samragh et al., 2023; Gromov et al., 2024; Kim et al., 2024; Song et al., 2024), with a few performance-based exceptions (Ma et al., 2023; Song et al., 2024). Motivated by the promising results of these algorithms, the more recent category of **Depth Pruning (Submodules)** further refined the granularity of the pruned structures, proposing both performance-based (Zhong et al., 2025) and similarity-based (Siddiqui et al., 2024; Sieberling et al., 2025) approaches.

## 3 Methods

We now introduce 2SSP, a novel pruning framework that combines two stages of structured pruning: a first stage (s1), which prunes at the level of entire neurons by removing rows and columns from the FFN within the Transformer blocks; and a second stage (s2), which performs submodule-based depth pruning by removing entire Attentions.

**Notations** Given a Transformer-based LLM $\mathcal{M}$, let $B$ denote the number of identical blocks it is made of, and $b$ represents a generic block. We define a *submodule* of block $b$ as either the Attention or the FFN of that block. The Attention mechanism involves three linear projections, namely queries $\mathbf{W}_{\text{query}}$, keys $\mathbf{W}_{\text{key}}$, and values $\mathbf{W}_{\text{value}}$, and an output projection $\mathbf{W}_{\text{out}}$, which for a given input $\mathbf{X}$ outputs a final representation computed as softmax$\left((\mathbf{X}\mathbf{W}_{\text{query}}(\mathbf{X}\mathbf{W}_{\text{key}})^\top)/\sqrt{d_k}\right)(\mathbf{X}\mathbf{W}_{\text{value}})\mathbf{W}_{\text{out}}$. For the most recent LLMs, such as the one tested in this paper, the FFN consists of three linear projections: gate $\mathbf{W}_{\text{gate}}$, up $\mathbf{W}_{\text{up}}$, and down $\mathbf{W}_{\text{down}}$, where the input is processed sequentially through the first two before the down projection. For a given input $X$, the FFN forward output is given by $(\sigma(\mathbf{X}\mathbf{W}_{\text{gate}})(\mathbf{X}\mathbf{W}_{\text{up}}))\mathbf{W}_{\text{down}}$, where $\sigma$ is the activation function.

The hidden dimension of the model is represented by $d_{\text{model}}$, while $d_{\text{int}}$ denotes the intermediate dimension of the FFN, corresponding to the output dimension of the gate and up projections, and to the input dimension of the down projection. For any Linear layer $l$ within these submodules, let $\mathbf{W}_l \in \mathbb{R}^{d_{\text{out}} \times d_{\text{in}}}$ denote the weight matrix of the layer. We use $\mathcal{D}_{\text{cal}}$ to denote the calibration dataset used for pruning.

**Neuron Pruning (s1)** The method aims to prune the neurons of the intermediate representation within the Linear layers of the FFN that follows the Attention in each Transformer block. The rationale is that the FFN's hidden state generates an intermediate representation that can be compressed by removing entire neurons, obtaining a lower-dimensional representation that preserves the most important features of the input sequence. We show that a high fraction of neurons in this representation created within the FFN is irrelevant, and thus can be pruned without significantly impacting performance. Moreover, our proposed neuron pruning scheme goes beyond simple magnitude-based pruning by aiming to *preserve the whole information path*, which has been theoretically shown to be one of the aspects correlated with better performance in sparse models Hoang et al. (2023); Cunegatti et al. (2024). In contrast to conventional approaches that simply eliminate

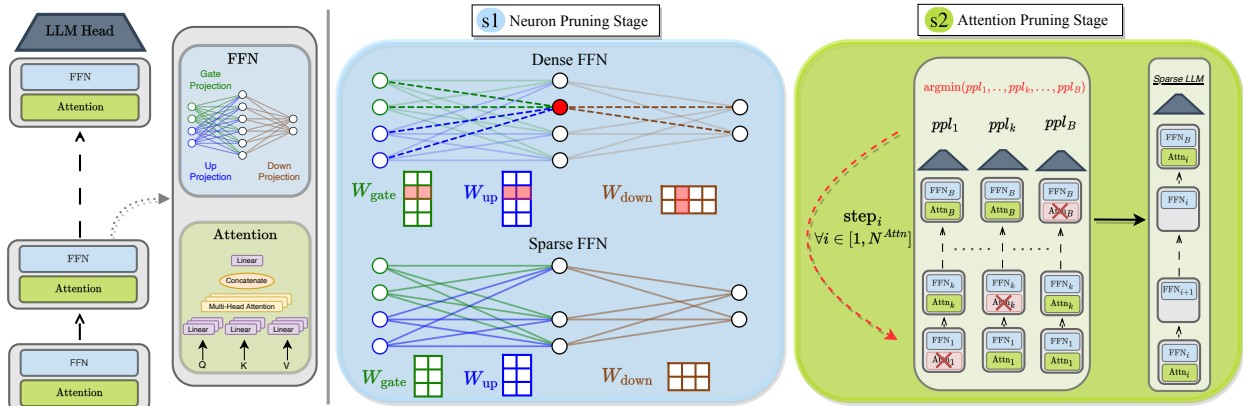

Figure 2: Conceptual scheme of the proposed 2SSP method. The left part of the image indicates the LLM given as input to the algorithm; the central part indicates stage s1, which focuses on FFNs; the right part indicates stage s2, which focuses on the Attention modules.

the highest magnitude weights, our method specifically targets the pruning of *pathways* connecting the three distinct FFN components (gate $\mathbf{W}_{\text{gate}}$, up $\mathbf{W}_{\text{up}}$, and down $\mathbf{W}_{\text{down}}$ projections).

Specifically, our algorithm prunes an equal number of neurons from each FFN, removing the neurons in the hidden state that have the lowest impact, measured as the magnitude of their activated output. The magnitude is calculated as the average $\mathcal{L}_2$ norm across the tokens in a set of calibration samples from a given calibration dataset $\mathcal{D}_{\text{cal}}$. The top-$K$ neurons are then retained in the FFN of each block.

Let the intermediate representation of the FFN in block $b$ be denoted as $\mathbf{Z}_b \in \mathbb{R}^{T \times d_{\text{int}}}$, where $T$ is the sequence length and $d_{\text{int}}$ is the dimension of the intermediate representation of the FFN in block $b$. For each neuron $j$, we compute an importance score $s_j$ across the calibration dataset $\mathcal{D}_{\text{cal}}$:

$$s_j = \frac{1}{|\mathcal{D}_{\text{cal}}|} \sum_{c=1}^{|\mathcal{D}_{\text{cal}}|} \|z_c^{(j)}\|_2 \tag{1}$$

where $z_c^{(j)}$ is the activation of the $j$-th neuron for the $c$-th sequence in the calibration dataset, and $\| \cdot \|_2$ denotes the $\mathcal{L}_2$ norm over the tokens in the calibration sequence.

Formally, let $\mathbf{W}_{\text{in}} \in \mathbb{R}^{d_{\text{int}} \times d_{\text{model}}}$ and $\mathbf{W}_{\text{out}} \in \mathbb{R}^{d_{\text{model}} \times d_{\text{int}}}$ be the input and output projection matrices of the FFN, respectively, where $d_{\text{model}}$ is the model's hidden dimension. We define a binary mask $\mathbf{m} \in \{0, 1\}^{d_{\text{int}}}$ that selects the top-$K$ neurons to preserve based on their importance scores $s_j$. The pruned weight matrices are then computed as:

$$\hat{\mathbf{W}}_{\text{in}} = \mathbf{W}_{\text{in}}[\mathbf{m} = 1, :] \tag{2}$$

$$\hat{\mathbf{W}}_{\text{out}} = \mathbf{W}_{\text{out}}[:, \mathbf{m} = 1] \tag{3}$$

To illustrate the mechanism, let us consider a simple FFN with two Linear layers, denoted as $l_{in}$ and $l_{out}$[1]. Removing a neuron from the hidden state of the FFN necessitates the removal of all the associated weights in both $l_{in}$ and $l_{out}$ connected to that particular neuron. More precisely, given the weight matrices $\mathbf{W}_{\text{in}} \in \mathbb{R}^{d_{\text{int}} \times d_{\text{model}}}$ and $\mathbf{W}_{\text{out}} \in \mathbb{R}^{d_{\text{model}} \times d_{\text{int}}}$, respectively of $l_{in}$ and $l_{out}$, the pruning of a hidden state neuron involves eliminating (in this order, to preserve network connectivity) the corresponding row in $\mathbf{W}_{\text{in}}$ and the associated column in $\mathbf{W}_{\text{out}}$.

**Attention Pruning (s2)** Pruning only the FFN parameters limits the effectiveness of the algorithm, as only a restricted fraction of the total number of parameters can be pruned, leaving all the Attention

---

[1]Without loss of generality, we represent the standard three-projections FFN (gate, up, down) into a two-layer representation where $l_{in}$ combines the gate and up projections, and $l_{out}$ handles the down-projection.

parameters intact. To address this limitation, we get inspiration from the observation derived in (Siddiqui et al., 2024), where it has been shown that Attentions can be removed to a certain degree ($\sim 33\%$) with almost no performance degradation, while the FFNs are more sensitive to depth removal. Based on this, we propose a second pruning stage that, after removing a certain fraction of FFN neurons in the first stage, applies depth pruning to Attentions[2]. We address this by adopting the submodules pruning mechanism proposed in (Zhong et al., 2025), by removing only Attentions (and not also FFNs as in the original mechanism) from the sparse network $\mathcal{M}_1$ obtained after the first stage. Specifically, we iteratively remove the Attentions leading to the lowest perplexity on the calibration dataset until the target sparsity is reached.

Formally, let $\mathcal{A} = \{a_1, \ldots, a_B\}$ be the set of Attentions across all the $B$ blocks of the Transformer. At each step $t$, we select the Attention module $a^*$ whose removal minimizes the perplexity on the calibration dataset:

$$a^* = \underset{a \in \mathcal{A}_t}{\operatorname{argmin}} \operatorname{PPL}(\mathcal{M}_t \setminus a, \mathcal{D}_{\text{cal}}) \tag{4}$$

where $\mathcal{A}_t$ is the set of the remaining Attention modules at step $t$, $\mathcal{M}_t \setminus a$ denotes the model at step $t$ with the Attention module $a$ removed, and PPL represents the perplexity metric, calculated as the exponential of the negated average log-likelihood over the calibration samples of $\mathcal{D}_{\text{cal}}$.

**Balancing the Sparsity Rate** As explained above, the proposed 2SSP algorithm works in two stages. Hence, given a target sparsity rate $s$, selecting the pruning rate allocated for each of the two stages is also required.

Through empirical analysis, we found out that the following equation provides a reliable metric for determining the optimal number of Attention modules to prune at any given sparsity rate (see Figure 6):

$$N^{\text{Attn}} = \operatorname{round}\left(B \cdot s^{\frac{|W_{\text{FFN}}|}{\alpha |W_{\text{Attn}}|}}\right) \tag{5}$$

where $\alpha = 1.5$ (see Section 4.3), $|W_{\text{FFN}}|$ represents the number of FFN parameters per block, and $|W_{\text{Attn}}|$ represents the number of Attention parameters per block. This equation captures two critical aspects of this pruning process. First, it ensures that the number of pruned Attention parameters scales with increasing sparsity rates. Second, it adjusts the Attention pruning rate based on the relative sizes of the FFN and Attention modules. Specifically, when $\frac{|W_{\text{FFN}}|}{|W_{\text{Attn}}|}$ is large, indicating that FFN parameters dominate the block structure, the equation reduces the proportion of Attention parameters to be pruned. This adaptive behavior helps maintain the balance between the number of Attentions and the number of FFN parameters pruned.

# 4 Experiments

**Setup** We implement 2SSP using PyTorch (Paszke et al., 2019) with HuggingFace's Transformer library (Wolf et al., 2020) for model and dataset management. All experiments are conducted on a cluster of four NVIDIA A30 GPUs (24GB memory each).

**Models** To test the effectiveness of our approach across different LLM families, we tested the 7B-parameter variants of Mistral-v0.3 (Jiang et al., 2023), Llama-2 (Touvron et al., 2023), and Qwen-2.5 (Yang et al., 2024), while using the 14B-parameter version Phi-3 (Abdin et al., 2024).

**Tasks and Datasets** We employ perplexity over language modeling tasks as the primary evaluation metric, given its established usage in assessing pruning algorithms (Frantar & Alistarh, 2023; Sun et al., 2024; Ashkboos et al., 2024; Sieberling et al., 2025) and the fact that it has been empirically proved to be a reliable metric for evaluating compressed models (Jin et al., 2024). We measure perplexity across three datasets: the full WikiText2 dataset (Merity et al., 2017), along with subsets of samples from C4 (Raffel et al., 2020) and FineWeb (Penedo et al., 2024) datasets, which are all popular benchmarks in the pruning literature (Sieberling et al., 2025). We also conduct downstream evaluations using LM Eval Harness (Gao et al., 2024),

---

[2]It should be noted that, in our method, width pruning is applied only to FFNs because, while information paths can be computed in FFNs (taking as neurons the activations of the hidden states), this is not possible for Attentions, because of the scaled dot-product between $\mathbf{K}$, $\mathbf{Q}$, and $\mathbf{V}$ of the different heads. This makes pruning of entire neurons impractical in Attentions.

including MMLU (Hendrycks et al., 2021), WinoGrande (WQ) (Sakaguchi et al., 2021), PiQA (Tata & Patel, 2003), HellaSwag (HS) (Zellers et al., 2019), and ARC (easy and challenge, in the following referred to as "ARC-e" and 'ARC-c", respectively) (Clark et al., 2018).

Table 1: Perplexity for 2SSP vs. the compared pruning algorithms over three different sparsity rates across four LLMs. The boldface and underline indicate, respectively, the best and second-best value per dataset (excluding the dense baseline).

| Sparsity | Method | Mistral-v0.3 7B | | | LLama-2 7B | | | Qwen-2.5 7B | | | Phi-3 14B | | |
|---|---|---|---|---|---|---|---|---|---|---|---|---|---|
| | | WikiText2 | C4 | Fineweb-Edu | WikiText2 | C4 | Fineweb-Edu | WikiText2 | C4 | Fineweb-Edu | WikiText2 | C4 | Fineweb-Edu |
| 0% | Dense | 5.36 | 8.13 | 6.49 | 5.47 | 7.13 | 6.44 | 6.85 | 11.68 | 7.7 2 | 4.31 | 8.54 | 6.41 |
| 25% | ShortGPT | 44.65 | 38.62 | 32.81 | 25.43 | 31.04 | 22.8 | 13.09 | 19.38 | 14.24 | 129.79 | 124.02 | 139.66 |
| | Sliding Window | 37.75 | 52.26 | 42.49 | 18.25 | 21.71 | 17.95 | $\underline{11.37}$ | 17.54 | 12.75 | 31.13 | 30.28 | 24.39 |
| | BlockPruner | 10.33 | 13.43 | 10.95 | 12.09 | 12.98 | 11.04 | 11.57 | 17.43 | 12.33 | 9.76 | 13.16 | 9.83 |
| | EvoPress | $\underline{9.35}$ | $\underline{12.86}$ | $\underline{10.39}$ | $\underline{10.37}$ | $\underline{11.92}$ | $\underline{10.23}$ | 11.74 | $\underline{17.29}$ | $\underline{12.41}$ | $\underline{8.71}$ | 12.34 | $\underline{9.53}$ |
| | SliceGPT | 11.84 | 13.09 | 11.36 | 14.82 | 12.57 | 11.19 | 12.72 | 17.40 | 13.88 | 10.01 | $\underline{11.86}$ | 9.82 |
| | 2SSP (Ours) | **9.24** | **12.19** | **10.27** | **9.25** | **10.52** | **9.21** | **10.61** | **15.67** | **11.92** | **7.06** | **10.43** | **8.42** |
| 37.5% | ShortGPT | $1.83e^3$ | $1.49e^3$ | $1.31e^3$ | 79.49 | 66.69 | 54.07 | 52.99 | 48.07 | 36.98 | $1.34e^5$ | $1.33e^5$ | $1.48e^5$ |
| | Sliding Window | 984.31 | $1.40e^3$ | $1.36e^3$ | 207.04 | 225.83 | 172.21 | 21.73 | 30.88 | 23.07 | $1.20e^6$ | $5.71e^5$ | $5.03e^5$ |
| | BlockPruner | 23.31 | 25.66 | 23.23 | 23.62 | 21.47 | 18.13 | 22.17 | 29.49 | $\underline{21.76}$ | 19.31 | 21.81 | 17.15 |
| | EvoPress | 27.00 | 24.10 | $\underline{19.93}$ | $\underline{19.03}$ | 20.22 | $\underline{17.04}$ | $\underline{22.03}$ | 28.98 | 21.79 | $\underline{15.62}$ | 19.89 | 15.18 |
| | SliceGPT | $\underline{23.24}$ | $\underline{21.41}$ | 19.98 | 30.28 | $\underline{19.58}$ | 18.71 | 25.08 | $\underline{28.07}$ | 25.31 | 17.06 | $\underline{15.81}$ | $\underline{14.52}$ |
| | 2SSP (Ours) | **14.92** | **17.15** | **15.03** | **14.64** | **14.93** | **13.36** | **15.26** | **20.95** | **16.89** | **9.79** | **12.88** | **10.91** |
| 50% | ShortGPT | $1.01e^3$ | 963.5 | 889.31 | 233.18 | 187.46 | 160.98 | $9.30e^4$ | $1.27e^8$ | $3.63e^8$ | $4.40e^5$ | $2.79e^5$ | $3.67e^5$ |
| | Sliding Window | $3.31e^3$ | $1.56e^3$ | $1.82e^3$ | $3.34e^3$ | $2.48e^3$ | $2.42e^3$ | 213.96 | 177.08 | 142.71 | $1.01e^6$ | $1.08e^6$ | $9.97e^5$ |
| | BlockPruner | 81.90 | 64.85 | 54.64 | 71.36 | 55.46 | 48.10 | 54.97 | 63.00 | 47.94 | 56.6 | 57.36 | 46.40 |
| | EvoPress | 91.83 | 73.71 | 60.55 | 70.97 | 44.39 | 38.58 | 49.91 | 57.95 | 43.45 | 54.46 | 50.22 | 39.95 |
| | SliceGPT | $\underline{41.24}$ | $\underline{35.29}$ | $\underline{33.96}$ | $\underline{57.66}$ | $\underline{36.06}$ | $\underline{35.55}$ | $\underline{38.47}$ | $\underline{41.98}$ | $\underline{37.56}$ | $\underline{34.40}$ | $\underline{26.42}$ | $\underline{27.21}$ |
| | 2SSP (Ours) | **23.77** | **25.95** | **23.30** | **31.40** | **27.16** | **25.40** | **21.66** | **28.00** | **23.72** | **16.93** | **18.82** | **17.07** |

**Baselines** We compare 2SSP against a broad set of state-of-the-art structured pruning methods for LLMs. Since our proposed approach works by combining width pruning with depth submodule pruning, we designed the experimental setup to include baselines from the three different categories of pruning granularity mentioned above:

• **Depth Pruning (Blocks)**: This category includes ShortGPT (Men et al., 2024) and Sliding Window Cosine Similarity (Gromov et al., 2024), which perform pruning at the coarsest level by removing entire Transformer blocks.

• **Depth Pruning (Submodules)**: This category includes BlockPruner (Zhong et al., 2025) and EvoPress (Sieberling et al., 2025), which shift the pruning target from entire Transformer blocks to Attention and/or FFN submodules.

• **Width Pruning**: This category includes SliceGPT (Ashkboos et al., 2024), which operates at the finest granularity by pruning individual rows and columns within Transformer weight matrices.

For all the pruning algorithms, as done for 2SSP, we use a calibration dataset made of sequences of 2048 tokens each, taken from the C4 dataset. For 2SSP, we use 32 sample sequences for the first stage (see Section 4.2) and one sample for the second stage. As done in the original papers, we set the number of samples for ShortGPT and Sliding Window Cosine Similarity to 256. For BlockPruner and EvoPress, we use a single calibration sample. Since these two methods are based on removing submodules and evaluating the solutions in an iterative manner (as our s2 stage), the number of samples highly influences the pruning runtime. For this reason, following the findings in (Sieberling et al., 2025), which demonstrate robust performance even with minimal calibration data, we set the calibration set size to one sample, as we have done for our second stage, aiming to obtain a fair comparison. For SliceGPT, we utilize 256 samples based on the empirical analysis showing performance convergence beyond 128 samples reported in (Ashkboos et al., 2024).

## 4.1 Experimental Results

In this section, we evaluate 2SSP w.r.t. the selected baselines in terms of performance over both language modeling and downstream tasks. We also include experiments about the runtime required by our approach to obtain the final sparse models. We show how our approach outperforms the competitors in terms of task performance and when confronting the pruning runtime vs. performance trade-off. We also evaluate the quality of the sparse models w.r.t. their inference speed-up w.r.t. the corresponding dense models.

**Numerical Results** The first task we evaluate is language modeling over WikiText2, C4, and FineWeb at three different sparsity rates, namely 25%, as in (Men et al., 2024; Zhong et al., 2025; Ashkboos et al., 2024), 37.5%[3], taken as intermediate value, and 50%, as in (Sieberling et al., 2025), avoiding higher values as it is established that structured pruning algorithms struggle to obtain reasonable performance above such sparsity rates.

Table 1 reports all the numerical results in terms of perplexity across the tested dataset and sparsity rates. It is clear how our proposed approach outperforms the baselines in all test cases. Even more interesting is the robustness of our approach w.r.t. both models and sparsity rates. Table 1 also highlights, for each setting, the second best (underline). Apart from the 50% case, where SliceGPT consistently ranks second, there are no baselines that outperform all the others after 2SSP. On the other hand, different baselines take the second place in different settings. In contrast, our approach provides stable results across all models, sparsity rates, and datasets tested.

We also tested if our approach could still outperform all the baselines at sparsity rates different from 25%, 37.5%, and 50%. For doing so, we computed the perplexity over WikiText2 for different sparsity rates ranging from zero to 70%. The results for Mistral-v0.3 7B and LLama-2 7B are shown in Figure 3, where the performance trend of 2SSP w.r.t. the baselines indicated in Table 1 is confirmed even across such a broader set of sparsity rates.

In order to assess the performance of our approach not only in terms of perplexity, we considered six different downstream tasks. In this case, we tested 2SSP, as well as the baselines, at 37.5% sparsity (the intermediate value among those considered), considering the average zero-shot and few-shot accuracy as the main metric. In detail, we conduct 3-shot evaluations on the same benchmark datasets from Table 2, namely MMLU (Hendrycks et al., 2021), WinoGrande (WQ) (Sakaguchi et al., 2021), PiQA (Tata & Patel, 2003), HellaSwag (HS) (Zellers et al., 2019), and ARC (easy and challenge, in the following referred to as "ARC-e" and 'ARC-c", respectively) (Clark et al., 2018). Table 2 shows the numerical results of the

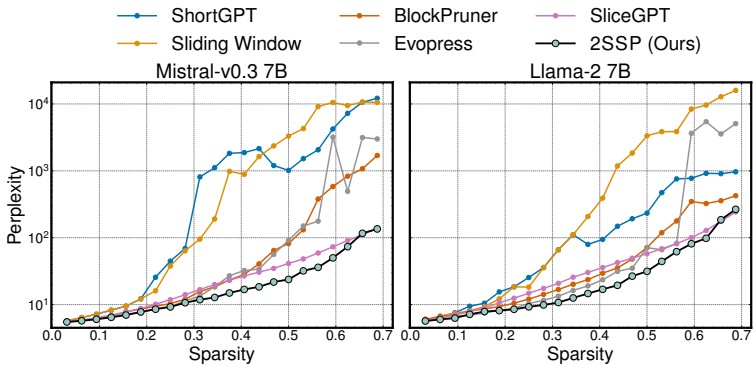

Figure 3: Perplexity for 2SSP vs. the compared pruning algorithms and the dense baseline when varying the sparsity rate on Llama-2 7B and Mistral-v0.3 7B.

zero-shot experimental setting, showing once again how 2SSP outperforms the tested baselines in most of the cases (and always on average across all tasks). Also in this case, no fixed second-best exists among the competitors, while our approach consistently outperforms (on average across all tasks) all the other methods, regardless of the model. Table 3 demonstrates once more that 2SSP consistently outperforms the baseline models across most tasks and maintains superior average performance in the few-shot (3-shot) regime across all models.

**Pruning Runtime** For any pruning algorithm to be considered effective, along with the performance obtained by the generated sparse model, also the time required to obtain such sparse models is a critical metric. For this reason, we compare the performance (in terms of perplexity over WikiText2) vs. the pruning runtime of 2SSP and the baselines. Figure 4 shows the trade-off between these two metrics for the three sparsity rates tested in Table 1, for the case LLama-2 7B (for the same analysis on the other models, please see the Appendix). The results clearly show how 2SSP is the best algorithm in terms of performance vs. pruning runtime trade-off. As expected, the fastest algorithms in terms of pruning runtime are the ones belonging to the *Depth Pruning (Blocks)* category, since they apply pruning in one-shot, as the first stage, w.r.t. the similarity among blocks. On the other hand, the algorithms from the *Depth Pruning (Submodules)*

---

[3]Given Qwen's architecture of 28 blocks, instead of 37.5% we select a sparsity rate of 39.29% as it represents the nearest achievable rate through whole-block pruning (i.e., 11 blocks removed).

Table 2: Zero-shot performance for 2SSP vs. the compared pruning models at 37.5% sparsity. The boldface and underline indicate, respectively, the best and second-best value per dataset (excluding the dense baseline).

| Model | Algorithm | MMLU | WQ | PIQA | HS | ARC-e | ARC-c | Average |
|---|---|---|---|---|---|---|---|---|
| Mistral-v0.3 7B | Dense | 59.08 | 73.72 | 80.30 | 60.91 | 79.67 | 48.81 | 67.08 |
| | ShortGPT | 22.67 | 58.56 | 56.96 | 27.73 | 33.59 | **29.27** | 38.13 |
| | Sliding Window | **25.54** | 57.22 | 59.19 | 29.41 | 34.51 | 26.88 | 38.79 |
| | BlockPruner | 23.59 | 54.62 | 66.16 | 37.92 | 46.25 | 24.32 | 42.14 |
| | EvoPress | 25.00 | 57.14 | 68.82 | 39.79 | 50.21 | 25.94 | 44.48 |
| | SliceGPT | 23.15 | 61.88 | 65.34 | 36.95 | 42.68 | 21.42 | 41.90 |
| | 2SSP | 24.49 | 63.14 | 70.29 | 41.99 | 49.96 | 24.49 | 45.73 |
| Llama-2 7B | Dense | 40.67 | 68.90 | 78.07 | 57.09 | 76.22 | 43.34 | 60.72 |
| | ShortGPT | 32.25 | 60.54 | 59.63 | 33.54 | 41.33 | **28.50** | 42.63 |
| | Sliding Window | **33.38** | 58.64 | 60.07 | 33.47 | 36.15 | 28.41 | 41.69 |
| | BlockPruner | 23.59 | 55.09 | 66.87 | 36.92 | 50.80 | 24.49 | 42.96 |
| | EvoPress | 25.66 | 52.01 | 68.61 | 37.15 | 53.20 | 25.94 | 43.76 |
| | SliceGPT | 23.07 | 63.85 | 67.90 | 40.40 | 47.56 | 26.19 | 44.83 |
| | 2SSP | 27.91 | 61.33 | 70.29 | 42.78 | 55.93 | 27.39 | 47.61 |
| Qwen-2.5 7B | Dense | 71.88 | 73.24 | 78.56 | 59.97 | 80.35 | 48.29 | 68.72 |
| | ShortGPT | 23.02 | 51.46 | 63.98 | 33.90 | 50.55 | 24.74 | 41.27 |
| | Sliding Window | 23.76 | 52.49 | 65.02 | 34.36 | 54.46 | 23.12 | 42.20 |
| | BlockPruner | 25.15 | 53.35 | 67.41 | 36.97 | 58.59 | 27.22 | 44.78 |
| | EvoPress | 24.21 | 55.17 | 67.30 | 36.95 | 59.76 | 27.13 | 45.09 |
| | SliceGPT | 22.91 | 57.70 | 65.94 | 34.32 | 48.02 | 20.48 | 41.56 |
| | 2SSP | 23.34 | 61.40 | 70.29 | 43.75 | 52.65 | 26.79 | 46.37 |
| Phi-3 14B | Dense | 67.61 | 75.77 | 81.01 | 64.04 | 84.05 | 60.67 | 72.19 |
| | ShortGPT | 27.03 | 52.09 | 56.04 | 27.15 | 34.34 | 28.50 | 37.53 |
| | Sliding Window | 25.17 | 50.04 | 52.77 | 25.65 | 26.22 | 23.04 | 33.82 |
| | BlockPruner | 27.90 | 61.40 | 68.12 | 42.00 | 62.75 | 37.37 | 49.93 |
| | EvoPress | 34.63 | 60.14 | 67.85 | 41.46 | 61.74 | 35.58 | 50.23 |
| | SliceGPT | 27.21 | 66.61 | 71.16 | 45.45 | 54.34 | 29.78 | 49.09 |
| | 2SSP | 51.85 | 68.82 | 74.97 | 51.60 | 67.26 | 38.99 | 58.92 |

Table 3: Few-shot performance (3-shot) for 2SSP vs. the compared pruning models at 37.5% sparsity. The boldface and underline indicate, respectively, the best and second-best value per dataset (excluding the dense baseline).

| Model | Algorithm | MMLU | WQ | PIQA | HS | ARC-e | ARC-c | Average |
|---|---|---|---|---|---|---|---|---|
| Mistral-v0.3 7B | Dense | 61.63 | 77.11 | 80.85 | 61.68 | 82.83 | 55.72 | 69.97 |
| | ShortGPT | 23.84 | 55.72 | 55.71 | 27.26 | 32.20 | **27.3** | 37.01 |
| | Sliding Window | 28.02 | 55.09 | 58.16 | 28.46 | 32.41 | 24.4 | 37.76 |
| | BlockPruner | 26.81 | 56.75 | 66.59 | 37.64 | 49.16 | 24.32 | 43.54 |
| | EvoPress | 25.37 | 57.14 | 68.12 | 39.62 | 51.43 | 27.05 | 44.79 |
| | SliceGPT | 27.12 | 60.69 | 62.95 | 36.13 | 42.09 | 21.50 | 41.75 |
| | 2SSP | 29.84 | 64.64 | 70.08 | 40.97 | 49.12 | 24.32 | 46.49 |
| Llama-2 7B | Dense | 45.65 | 71.98 | 78.35 | 57.95 | 79.21 | 47.87 | 63.50 |
| | ShortGPT | 39.16 | 62.59 | 60.34 | 33.86 | 43.73 | 29.86 | 44.92 |
| | Sliding Window | 39.27 | 58.64 | 58.32 | 32.42 | 38.01 | 29.18 | 42.64 |
| | BlockPruner | 24.78 | 54.54 | 66.59 | 36.90 | 52.78 | 26.02 | 43.60 |
| | EvoPress | 25.54 | 51.93 | 68.17 | 36.80 | 54.63 | 27.13 | 44.03 |
| | SliceGPT | 27.03 | 62.90 | 67.41 | 39.28 | 49.16 | 27.47 | 45.54 |
| | 2SSP | 28.38 | 59.27 | 70.08 | 41.00 | 57.15 | 29.78 | 47.61 |
| Qwen-2.5 7B | Dense | 74.01 | 74.74 | 80.36 | 59.76 | 86.07 | 58.79 | 72.29 |
| | ShortGPT | 24.92 | 50.67 | 64.2 | 34.14 | 53.96 | 25.85 | 42.29 |
| | Sliding Window | 26.58 | 50.67 | 64.04 | 34.12 | 54.00 | 22.95 | 42.06 |
| | BlockPruner | 26.09 | 53.43 | 67.30 | 36.19 | 59.39 | 29.01 | 45.23 |
| | EvoPress | 26.63 | 52.49 | 66.97 | 36.83 | 60.27 | 27.05 | 45.04 |
| | SliceGPT | 27.00 | 55.25 | 63.82 | 32.91 | 44.36 | 19.80 | 40.52 |
| | 2SSP | 37.44 | 62.51 | 70.29 | 42.99 | 52.57 | 27.82 | 48.94 |
| Phi-3 14B | Dense | 76.51 | 73.88 | 81.12 | 64.84 | 87.25 | 62.37 | 74.33 |
| | ShortGPT | 36.38 | 55.17 | 57.78 | 27.33 | 38.22 | 29.01 | 40.65 |
| | Sliding Window | 24.21 | 51.62 | 52.39 | 25.77 | 26.05 | 22.95 | 33.83 |
| | BlockPruner | 37.20 | 58.33 | 68.82 | 42.06 | 63.09 | 37.71 | 51.20 |
| | EvoPress | 35.47 | 61.96 | 70.13 | 41.69 | 65.99 | 37.88 | 52.19 |
| | SliceGPT | 37.24 | 69.69 | 71.22 | 45.30 | 56.14 | 30.63 | 51.70 |
| | 2SSP | 52.08 | 70.01 | 75.73 | 51.47 | 68.60 | 40.27 | 59.70 |

category are the slowest ones, since they evaluate each possible submodule removal combination. The pruning runtime of SliceGPT lies somewhere in between and is mostly due to the PCA step employed for reducing the matrix dimension. To conclude, our approach, which is a combination of one-shot removal ( s1 ) and submodule evaluation ( s2 ), is able to achieve the best performance in terms of perplexity while requiring limited pruning runtime.

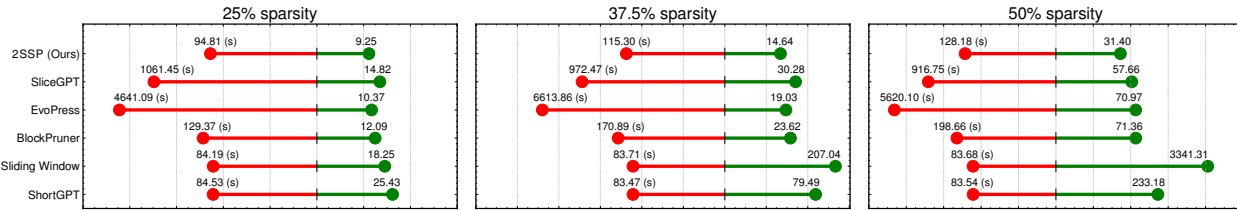

Figure 4: Pruning runtime (red, left side of the x-axis, log scale) vs. perplexity (green, right side of the x-axis, log scale) for 2SSP vs. the compared pruning algorithms over LLama-2 7B pruned at three different sparsity rates.

**Sparse Inference Speed-Up** To complete the analysis of 2SSP and the baselines, we study the inference runtime speed-up of the sparse models generated with the different pruning algorithms under examination over three different NVIDIA GPU models. Table 4 reports the inference runtime of LLama-2 7B (including its dense version) computed as the average inference GPU time (in ms) over 10 runs on a single sample of 2048 tokens. The speed-up of the structured pruning algorithms w.r.t. the dense model is evident and, as expected, increases with the sparsity rate. The advantage is maximum for the *depth* pruning approaches, while the *width* pruning methods achieve minimum speed-ups. As expected, 2SSP positions itself between the two categories.

Table 4: Inference time (ms) of pruned models on different GPUs.

| GPU | Sparsity | Pruning Method | | | | | |
|-----|----------|----------|----------------|-------------|----------|----------|------|
| | | ShortGPT | Sliding Window | BlockPruner | Evopress | SliceGPT | 2SSP |
| **RTX 3090 24GB** | Dense | | | 457.59 | | | |
| | 25% | 375.86 | 362.25 | 357.31 | 352.95 | 410.33 | 381.22 |
| | 37.5% | 318.26 | 314.61 | 301.84 | 297.03 | 330.92 | 316.66 |
| | 50% | 258.46 | 253.46 | 245.30 | 242.01 | 286.33 | 270.82 |
| **A30 24GB** | Dense | | | 317.39 | | | |
| | 25% | 241.85 | 241.82 | 243.45 | 242.47 | 269.56 | 255.09 |
| | 37.5% | 203.07 | 203.66 | 204.80 | 204.49 | 228.48 | 223.41 |
| | 50% | 163.96 | 163.86 | 167.15 | 166.65 | 206.30 | 186.92 |
| **A100 80GB** | Dense | | | 158.17 | | | |
| | 25% | 120.49 | 120.48 | 120.21 | 121.43 | 129.17 | 127.53 |
| | 37.5% | 100.44 | 100.94 | 100.92 | 102.09 | 113.47 | 111.11 |
| | 50% | 81.22 | 81.39 | 81.40 | 83.09 | 99.17 | 92.59 |

## 4.2 Ablation Studies

In this section, we conduct different ablation studies to test the robustness of our proposed approach. In particular, we focus our analysis on the first stage ( s1 ) and on the combination of depth pruning and width pruning, which are the distinctive aspects of our method.

**Pruning Rows-Columns vs. Columns-Rows in s1** We conduct an ablation study to analyze the first stage of 2SSP, hence the neuron-based pruning approach. As discussed earlier, given the intermediate representation of an FFN, our method removes entire neurons by pruning their corresponding rows and

columns in the input and output projection matrices, respectively. We explore an "inverted" approach that, instead, prunes columns in the input projection matrix and rows in the output projection matrix, thereby preserving all neurons in the intermediate representation while reducing the dimensionality $d_{\text{model}}$ of the hidden state. The perplexity results in Table 5 show how the row-columns strategy employed by 2SSP is the most reliable choice since inverting the order or pruning leads to worse results even by orders of magnitude in terms of perplexity. This can be explained by the fact that pruning rows first and then columns (but not the other way around) preserves the network connectivity.

**Neuron Selection based on $\mathcal{L}_1$ vs. $\mathcal{L}_2$ in s1** To evaluate the robustness of our neuron selection criterion, we study the impact of using $\mathcal{L}_1$ norm as an alternative magnitude metric for neuron selection, comparing it against the $\mathcal{L}_2$ norm used in Eq. 1. In this variation, we modify the importance score $s_j$, by replacing the $\mathcal{L}_2$ norm with the $\mathcal{L}_1$ norm. The perplexity results in Table 5 show that using $\mathcal{L}_1$ leads to worse performance across all models and sparsity rates compared to the main 2SSP version based on $\mathcal{L}_2$.

**Running s1 only vs. s1 + s2**

We also separate the two stages included in 2SSP, to show the effectiveness of their combination. However, while it is possible to ablate on the first stage only, we exclude the case of having the second stage only since this stage's applicability is constrained by the model's architecture, hence it is not applicable to all models. In fact, the number of Attention parameters in an LLM ranges from approximately 19% in Llama-2 to 33% in Mistral-v0.3, which limits the possibility of conducting this ablation across the three different sparsity rates chosen throughout the whole paper.

On the other hand, this does not hold for the first stage alone, for which results are reported in Table 5. An interesting trend emerges in this case: while 2SSP (combining both stages) always outperforms the first stage only, it is visible how the gap between these two approaches increases while increasing the sparsity rate. It should be noted, however, that also the first stage alone cannot be applied to extreme sparsity rates such as $> \sim 70\%$ due to the same limitations of the second stage only, in this case related to the number of FFN parameters.

Table 5: Numerical results of the ablation studies over Mistral-v0.3 7B and LLama-2 7B for three different sparsity rates. The results correspond to the perplexity computed over the WikiText2 dataset.

| Model | 2SSP variant | Sparsity | | |
|---|---|---|---|---|
| | | 25% | 37.5% | 50% |
| **Mistral-v0.3 7B** | s1 inverted + s2 | 257.30 | $1.23e^3$ | $5.80e^3$ |
| | s1 $\mathcal{L}_1$ norm + s2 | 9.41 | 17.65 | 48.52 |
| | s1 only | 9.51 | 15.58 | 30.16 |
| | s1 + s2 | **9.24** | **14.92** | **23.77** |
| **LLama-2 7B** | s1 inverted | $8.27e^3$ | $1.02e^4$ | $5.51e^4$ |
| | s1 $\mathcal{L}_1$ norm | 11.37 | 23.64 | 66.56 |
| | s1 only | 9.88 | 19.94 | 41.73 |
| | s1 + s2 | **9.19** | **18.16** | **32.85** |

## 4.3 Hyperparameter Tuning

2SSP relies on two main hyperparameters, namely $\alpha$ used in Eq. 5 and the calibration set size $|\mathcal{D}_{\text{cal}}|$. We detail their selection choices in this section. Note that for the calibration set size of the second stage, we set it to one sample, as done in (Sieberling et al., 2025), to balance pruning runtime vs. performance and provide a fair comparison, as also explained in Section 4.1. Therefore, we limit our analysis to the calibration set size applied to the first stage.

**Choice of $\alpha$ parameter** A crucial aspect that makes our method effective is an accurate balancing between the number of parameters pruned in the first and the second stages. To achieve this balance, we introduced Eq. 5, which, given a desired sparsity rate $s$ for the whole model, controls the number of Attentions to prune ($N^{\text{Attn}}$). This equation uses a parameter $\alpha$ to regulate the rate of Attention pruning w.r.t. the overall sparsity: lower values of $\alpha$ result in a more gradual increase in the number of Attentions w.r.t. $s$, while higher values of $\alpha$ produce higher values of $N^{\text{Attn}}$ already at lower sparsity rates. Through empirical evaluation across multiple sparsity rates and models, we determine $\alpha = 1.5$ to be near-optimal, as shown in Figure 5.

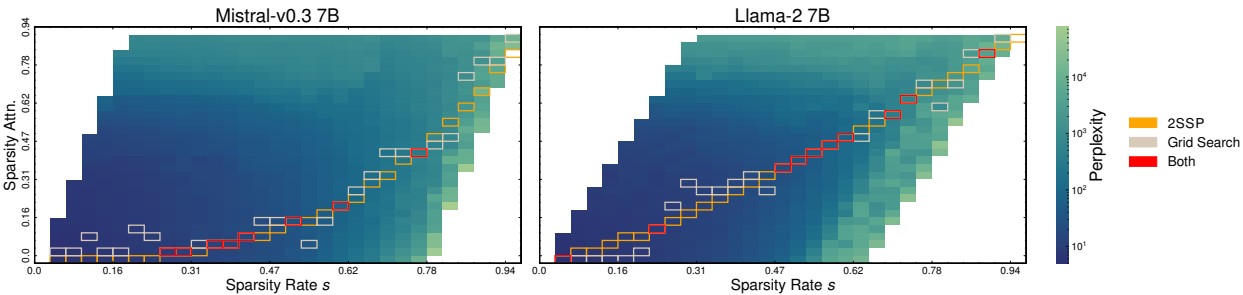

Figure 6: Perplexity on WikiText2 for various model sparsities (steps of 0.03) and Attention sparsities (pruned/total Attentions). The cells with orange (gray) border indicate the optimal no. of Attention modules to prune for each FFN sparsity, obtained by grid search (by 2SSP with Eq. 5). The red border indicates that the two methods find the same value. Eq. 5 is a valid proxy for the optimal value. The white cells correspond to sparsity values where 2SSP cannot be applied due to the theoretical maximum sparsity constraint.

Figure 6 shows, for Mistral-v0.3 7B and Llama-2 7B, the perplexity across various combinations of overall sparsity rates ($s$) and Attention sparsities. The figure highlights the importance of our proposed sparsity balancing approach. When pruning only FFNs (i.e., when the Attention sparsity is zero), perplexity deteriorates rapidly at higher sparsity rates, particularly when approaching the theoretical maximum sparsity of $\frac{B \cdot |W_{\text{FFN}}|}{|W_{\text{total}}|}$. This limit is imposed by the total number of FFN parameters available for pruning. However, we observe that gradually incorporating Attention pruning alongside neuron pruning leads to improved perplexity scores. The figure also shows the robustness of our $\alpha$ selection choice. Comparing the value of $N^{\text{Attn}}$ found by Eq. 5 setting $\alpha = 1.5$ with the optimal value found by an exhaustive grid search spanning all possible choices of Attention sparsity, it can be seen there is in most

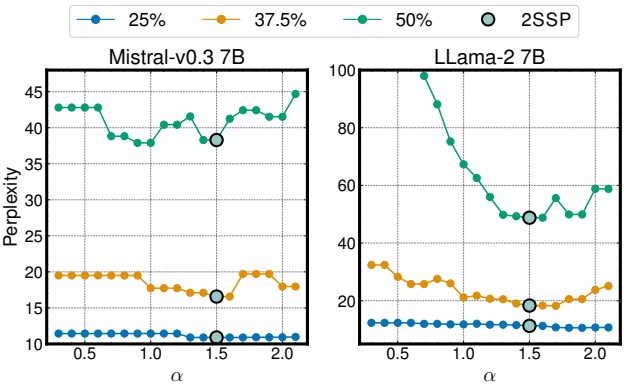

Figure 5: Perplexity on WikiText2 of models pruned using 2SSP at different sparsity rates when varying $\alpha$ in Eq. 5.

cases an almost exact match. It is also noteworthy to observe that the optimal number of Attention parameters to prune is highly dependent on the model; nevertheless, our approach reliably provides a near-optimal choice.

**Choice of Calibration Set Size** Finally, we investigate how the number of calibration samples for s1 affects model performance while maintaining a fixed sequence length of 2048 tokens. We conduct this analysis on Wikitext2 with both Mistral-v0.3 and Llama-2 at 50% sparsity, using perplexity to evaluate the performance of the final pruned models after applying 2SSP with varying calibration set sizes. The results shown in Figure 7 reveal that 2SSP achieves strong performance even with a limited number of calibration samples (at least 16). Given that larger calibration sets increase computational overhead due to additional forward passes, we stick to 32 calibration samples for the main experiments, as mentioned earlier in Section 4.1.

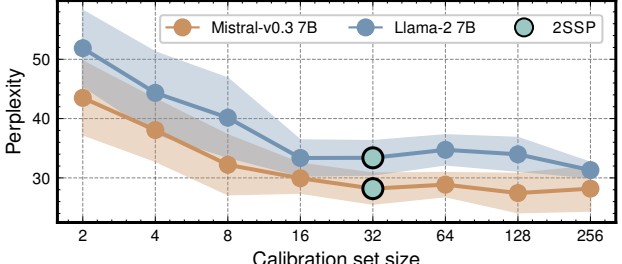

Figure 7: Perplexity on WikiText2 of models pruned using 2SSP at 50% sparsity when varying the calibration set size.

## 5    Conclusions

In this paper, we introduced 2SSP, a new structured pruning algorithm that aims to combine *Width Pruning* for FFNs with *Depth Pruning* for Attentions. Our approach works in two stages by firstly pruning neurons in the intermediate state of FFNs, and then iteratively removing Attentions based on the model performance computed as perplexity. We tested 2SSP over three different families of LLMs, ranging from 7B to 14B at three different sparsity rates. The results demonstrate how our proposed algorithm consistently outperforms the state-of-the-art baselines on both language modeling and downstream tasks. 2SSP achieves these results while requiring limited pruning runtime, which positions our method as state-of-the-art over the performance vs. pruning runtime trade-off. Finally, we conducted in-depth ablation and tuning studies to demonstrate the robustness of our proposed method, focusing in particular on the neuron pruning mechanism of the first stage and the sparsity rate balancing between the two stages.

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

# A Details on Pruning Rows-Columns vs. Columns-Rows in s1

For each dimension $k$ in the inputs' and outputs' hidden state, we compute importance scores across the calibration dataset $\mathcal{D}_{\text{cal}}$:

$$s_k^{\text{in}} = \frac{1}{|\mathcal{D}_{\text{cal}}|} \sum_{c=1}^{|\mathcal{D}_{\text{cal}}|} \|i_c^{(k)}\|_2 \tag{6}$$

$$s_k^{\text{out}} = \frac{1}{|\mathcal{D}_{\text{cal}}|} \sum_{c=1}^{|\mathcal{D}_{\text{cal}}|} \|o_c^{(k)}\|_2 \tag{7}$$

where $i_c^{(k)}$ and $o_c^{(k)}$ denote the $k$-th dimension of the input and output hidden states respectively. We define binary masks $\mathbf{m}_{\text{in}}, \mathbf{m}_{\text{out}} \in \{0,1\}^{d_{\text{model}}}$ to select the top-K dimensions based on these importance scores. The pruned weight matrices are computed as:

$$\hat{\mathbf{W}}_{\text{in}} = \mathbf{W}_{\text{in}}[:, \mathbf{m}_{\text{in}} = 1] \tag{8}$$

$$\hat{\mathbf{W}}_{\text{out}} = \mathbf{W}_{\text{out}}[\mathbf{m}_{\text{out}} = 1, :] \tag{9}$$

We evaluate both pruning strategies on the WikiText2 dataset across various sparsity rates as shown in Table 5. Our experimental results demonstrate that pruning entire neurons consistently outperforms the dimension-based pruning approach in terms of perplexity.

# B Additional Results

## B.1 Performance vs. Pruning Runtime

Here we present additional results of performance (perplexity on WikiText2) vs. pruning runtime of 2SSP and the baselines on Mistral-v0.3 7B (Figure 8), Qwen-2.5 7B (Figure 9), and Phi-3 14B ( Figure 10). Across all models and sparsity rates, our 2SSP consistently demonstrates superior performance compared to existing pruning methods.

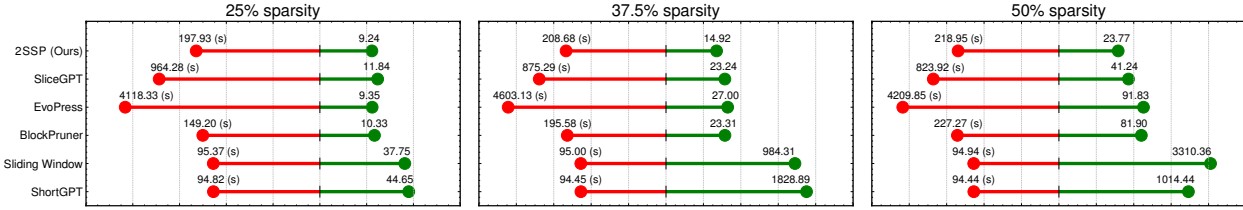

Figure 8: Pruning runtime (left side of the x-axis, log scale) vs. perplexity (right side of the x-axis, log scale) for 2SSP vs. the compared pruning algorithms over Mistral-v0.3 7B pruned at three different sparsity rates.

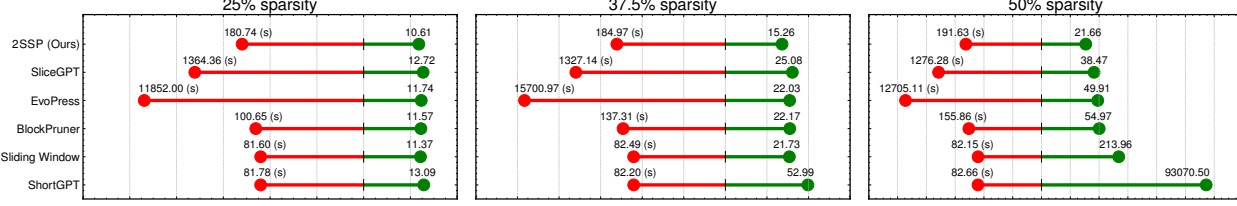

Figure 9: Pruning runtime (left side of the x-axis, log scale) vs. perplexity (right side of the x-axis, log scale) for 2SSP vs. the compared pruning algorithms over Qwen-2.5 7B pruned at three different sparsity rates.

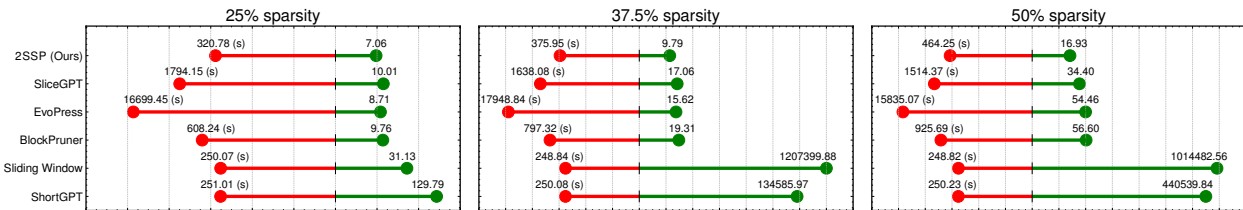

Figure 10: Pruning runtime (left side of the x-axis, log scale) vs. perplexity (right side of the x-axis, log scale) for 2SSP vs. the compared pruning algorithms over Phi-3 14B pruned at three different sparsity rates.

## C   Experimental Details

We implement our experiments using the HuggingFace Transformers library (Wolf et al., 2020). The models we use are sourced from the HuggingFace model hub, with their corresponding repositories listed in Table 6.

| Model | HuggingFace Repository |
|---|---|
| Mistral-v0.3 7B | `mistralai/Mistral-7B-v0.3` |
| Llama-2 7B | `meta-llama/Llama-2-7b-hf` |
| Qwen-2.5 7B | `Qwen/Qwen2.5-7B` |
| Phi-3 14B | `microsoft/Phi-3-medium-128k-instruct` |

Table 6: Models and their corresponding HuggingFace repositories.

For the calibration dataset, we use the Colossal Clean Crawled Corpus (C4) (Raffel et al., 2020), specifically the HuggingFace repository `allenai/c4`. Due to the substantial size of the training split, we fetch only a subset of samples extracted from the first shard of the dataset. Since sequence lengths in C4 vary, we first concatenated all sequences into a single corpus, tokenized it, and then divided it into segments of exactly 2048 tokens each.

For evaluation, we use the `wikitext-2-raw-v1` split available from the `wikitext` repository on HuggingFace. Additionally, for perplexity evaluation, we include two other datasets: a subset of the validation split of C4 and samples from the `sample-10BT` subset of the `HuggingFaceFW/fineweb-edu` repository. Similar to the calibration dataset, the evaluation datasets are processed by concatenating their sequences, tokenizing the resulting corpus, and splitting it into sequences of 2048 tokens each. Specifically, Wikitext consists of 163 sequences, C4 contains 288 sequences, and FineWeb comprises 259 sequences.

For reproducibility, we set a fixed random seed of 0 across all randomization sources in our experiments by enabling deterministic mode for CUDA operations and setting the seed for Python's random module, NumPy, and PyTorch. This ensures consistent results across different runs of our experiments.

## D   Qualitative Results

In this section, we present some qualitative examples obtained from pruned models by evaluating their performance on concrete generation tasks. Specifically, we examine the Mistral-v0.3 model pruned using 2SSP at three distinct sparsity rates: 25%, 37.5%, and 50%. The pruned models are tested on the content generation task starting from two distinct prompts: "Who is Albert Einstein?" and "Explain the theory of relativity". We compare the quality of their generated responses at each sparsity rate, providing examples for each case. The results for the first prompt are shown in Table 7, while those for the second prompt are shown in Table 8.

The qualitative results confirm that, as expected, higher sparsity leads to a decrease in the quality of generated responses. However, it is noteworthy that despite the reduced accuracy, 2SSP is capable of maintaining the structural coherence of the sentences, often preserving the flow of the narrative. This coherence is

achieved at the cost of factual correctness, with pruned models occasionally introducing incorrect or irrelevant information.

Table 7: Qualitative examples of text generated by Mistral-v0.3 7B pruned at different sparsity rates with 2SSP. The responses are based on the prompt "Who is Albert Einstein?".

| Sparsity | Generated Sentences |
|---|---|
| Dense | Albert Einstein was a German-born theoretical physicist who developed the theory of relativity, one of the two pillars of modern physics (alongside quantum mechanics). His work is also known for its influence on the philosophy of science. He is best known for his mass–energy equivalence formula E = mc2 (which has been dubbed "the world's most famous equation"). He received the 1921 Nobel Prize in Physics for his services to theoretical physics, and especially for his discovery of the law of the photoelectric effect |
| 25% | Albert Einstein was born on January 1, 1879, in Switzerland. He was a physicist and won the Nobel Prize in 1991. He is best known for his theory of relativity. Einstein was born in Switzerland, but he grew up in Germany. He studied at the University of Zurich, where he earned his doctorate in physics. He then went on to work at the Swiss Federal Institute of Technology in Zurich. There, he worked on a number of projects, including the development of the atomic bomb. |
| 37.5% | Albert Einstein was born on January 21, 1905, and died on April 28, 2021. He was a Nobel Nobel Prize winner and the winner of Nobel's Nobel Award for Nobel Science. Einstein was born in the United States, and he was the first Nobel to win Nobel and Nobel awards. He won Nobel in Nobel, and his Nobel award was Nobel. He also received Nobel Awards for Noble Science and Noble Noble Awards. Albert Einstein |
| 50% | Albert Einstein was born in 1990 in the United States. Albert Einstein is a well-known artist and sculptor. He is famous for his sculpting skills. He has been sculpting since 2010. Albert is a sculptor by profession. ## Albert Einstein Biography He was born on the 17th of July, 1890, in the USA. Albert has been working in the sculpting industry since he was 10 years old. He started sculpting at the age of 15. |

Table 8: Qualitative examples of text generated by Mistral-v0.3 7B pruned at different sparsity rates with 2SSP. The responses are based on the prompt "Explain the theory of relativity".

| Sparsity | Generated Sentences |
|---|---|
| Dense | The theory of relativity is one of the most important theories in the history of science. It was developed by Albert Einstein in 1905 and published in his paper "On the Electrodynamics of Moving Bodies." In this paper, Einstein proposed that the laws of physics are the same for all observers, regardless of their state of motion. This theory has revolutionized our understanding of the universe and has had a profound impact on many fields of science, including physics, astronomy, and cosmology. One of the key concepts in the theory of relativistic is the concept of spacetime. |
| 25% | The theory of relativity is one of the most important theories in physics. It is based on the idea that the speed of light is the same for all observers, regardless of their distance from each other. This means that no matter how far apart two people are, they will still be able to see each other at the same time. The theory was first proposed by Albert Einstein in 1905, and it has since been used to explain many different phenomena. For example, it can explain why some objects appear to be moving faster than others, even though they are not actually moving at all. |
| 37.5% | The theory of relativity is one of the most important concepts in the world of physics. It is a concept that has been around since the beginning of the 20th century, and it has been used to explain how the universe works. The theory of physics is a very important concept in physics. Physics is a field of physics that deals with the physics of the universe. It deals with physics, physics, and physics. The physics of physics are very important because they help us to understand the universe better. Physics is very important in physics because it helps us to better understand physics. For example, physics |
| 50% | The theory of relativity in the United States of America (US) has been the topic of discussion for a long time now. The US is the only country in the world that has a reputation for being the best in the US. The US is a country that has been in existence since the 1950s. It is the US that has the reputation of being one of the most popular countries to live in. This is because of the US' reputation for the US to be a country where the US can be found to be the best country to be in. In this article, we will discuss the US |

