# OpenReview forum: "2SSP: A Two-Stage Framework for Structured Pruning of LLMs"
_TMLR — Accepted by TMLR_

### Review · Reviewer_Ngzt · 2025-05-22

**Summary Of Contributions:**

This paper introduces 2SSP, a two-stage structured pruning framework that combines width and depth pruning strategies for LLM. The key contributions are:
- Two-stage pruning framework: Stage 1 (width pruning) removes whole neurons in the FFN based on L2 norm importance scores. Stage 2 (depth pruning) iteratively removes attention modules that minimize perplexity impact.
- Sparsity balancing mechanism: A sparsity-balancing rule (Eq. 5) automatically allocates how much to prune in each stage.
- Comprehensive evaluation: Testing across four LLM families (Mistral, Llama-2, Qwen-2.5, Phi-3) at multiple sparsity rates (25%, 37.5%, 50%) on both language modeling and downstream tasks.
- Useful ablation study: Ablations justify the row-then-column order, the L2 importance metric, and the need to combine both stages﻿.

**Audience:**

Yes

**Broader Impact Concerns:**

The paper does not raise major ethical red flags. However, there may always be concerns about the fairness, security, and hallucination issues for a pruned LLM. While I do not think it is always necessary to add this broader impact statement, it would be welcomed if authors were willing to discuss it.

**Claims And Evidence:**

Yes

**Requested Changes:**

I do not find any changes that seem to be **critical** for this paper's acceptance. However, there are a few optional changes that can increase the chance of acceptance including: a) fairness of experiments about the calibration set size, b) some theoretical analysis, c) boarded sparsity exploration (as explained above), d) a broader impact statement section (as explained below).

**Strengths And Weaknesses:**

### Strengths:
- Novelty: Interesting idea to combine width and depth pruning strategies.
- Solid performance:  Comprehensive evaluation across multiple models, datasets, and metrics with proper baseline comparisons.
- Practical efficiency: Achieves strong performance with reasonable computational overhead, balancing one-shot removal (s1) with iterative evaluation (s2).
- Extensive ablation study: Extensive ablation studies examining neuron selection criteria, pruning order, hyperparameter sensitivity, and calibration set size.
- Nice presentation and clarity: Clear algorithmic description, simple importance metric, analytically motivated balancing rule.

### Weaknesses:
- Fairness: The authors utilize an ablation study to choose the size of the calibration set (Fig. 7). However, for the rest of the baselines, the authors use different calibration set sizes for different baselines. While authors did provide rationale for why they do so (For BlockPruner and EvoPress the authors trade away calibration data to hold wall-clock time in check, whereas for SliceGPT they trade in more data to maximise accuracy once its curve “converges beyond 128.”), It mixes two objectives (speed vs. quality) across baselines. It would be good to give both views: (a) accuracy-max settings (each method’s recommended sample count, even if slower, like for BlockPruner, the original setting is 256) and (b) matched-runtime settings (few samples for all). Readers can then see the speed/accuracy frontier.
- Limited theoretical analysis: While the idea is interesting, the paper lacks theoretical justification for why combining width and depth pruning should work better than individual approaches. Adding this would provide interesting insights to the readers.
- Limited sparsity exploration: As acknowledged by the authors themselves, due to limitations of the methodology, the maximal sparsity that can be achieved is 70%, which may not be satisfied in certain usage scenarios. Even so, the maximum presented sparsity in this paper is 50%, which is even lower.

---

> ### Author Response · Authors · 2025-06-17
>
> We thank the reviewer for the valuable feedback. Below are our comments related to the Weaknesses.
>
> **W1 - Fairness**: Motivated by this reviewer’s comment, we further analyzed the original results of BlockPruner [7] and EvoPress [4], which are the methods for which the calibration data size has been changed in our experiments (w.r.t. the setting recommended in their original papers) for consistency with our method.
>
> In the original BlockPruner paper [7], using 256 calibration samples, the authors reported a perplexity of **11.51** over Llama2 7B evaluated on Wikitext2 at 22% sparsity, while, in our experimental setup, BlockPruner resulted in **12.37** perplexity using only 1 sample, on the same dataset, and sparsity rate. When comparing these results with those achieved by 2SSP on the same model, dataset, and sparsity level, our method achieves a perplexity of **9.31** using 1 sample (for the depth pruning step). Note that the 22% sparsity results are not reported in our paper, but we ran the experiment only for the sake of the comparison reported in this rebuttal.
>
> Concerning EvoPress, the perplexity reported in the original paper [4] (again, on WikiText-2 over LLama-2 7B) has been obtained using 65536 tokens ($\sim$ 32 samples of 2048 tokens) and is respectively: for 25% sparsity, **9.15**; for 37,5% sparsity, **17.98**; and, for 50%, **48.8**. The perplexity results of EvoPress reported in our paper, using only 1 sample, with the same model, dataset and sparsity levels, are respectively **10.37**, **19.03**, and **57.66**, which are in fact not too far from the original ones (despite the reduced pruning time). To note that the perplexity results on the same model, dataset and sparsity levels obtained with 2SSP with 1 sample (for the depth pruning step) are respectively **9.25**, **14.64**, and **31.40**, which are better (apart from the 25% case) than the original results presented in the EvoPress paper [4].
>
> Overall, while we agree with the reviewer that it would have been definitely interesting to analyze in detail the trade-off between accuracy-max and runtime, the high computational cost required for running those algorithms with various values of calibration set sizes (especially for large values) unfortunately goes beyond our availability of GPUs resources. We hope that the comparison reported here in the rebuttal is sufficient to address this reviewer’s suggestion.
>
> **W2 - Limited Theoretical Analysis**: First of all, we agree with the reviewer that our paper does not propose any novel theoretical insights w.r.t. the LLM compression field. On the other hand, our proposed combination of depth and width pruning is grounded in recent works, both empirical and theoretical, that justify such a combination. The first justification relies on the empirical analysis done in [1], and the further confirmed by our experiments in Figure 6, where it is shown how, for the case of depth pruning, Attention submodules are highly redundant: in fact, removing up to one third of these submodules results in minimal degradation in performance compared to the dense model. On the other hand, width pruning approaches (such as SliceGPT) provide better task performance (perplexity and/or accuracy), due to the higher granularity of their corresponding search space [1]. Therefore, our proposed 2SSP intends to integrate the benefits of both depth and width pruning in a single method.
>
> Moreover, the rationale behind our proposed neuron pruning scheme is that not only it allows to perform pruning based on magnitude, but it also aims to *preserve the whole information path*, which has already been theoretically shown to be one of the aspects correlated with better performance in sparse models [5, 6]. In fact, in our method, we don’t simply prune the highest magnitude weights, but we focus on pruning *the paths* between the three different components (Gate, Up, and Down projections) of the FeedForward.
>
> **W3 - Limited sparsity exploration**: We would like to point out that, in Figure 3 (p. 7), the experimental results are available for a broader range of sparsity levels, up to 70%. The results show the perplexity over WikiText-2 for 2SSP and the baselines. As mentioned in the paper, while in principle our approach could be applied for sparsity levels even higher than 70%, we purposely avoided to test those values since such high sparsity levels are rarely considered in structured pruning research (due to the anticipated high gap in performance w.r.t. the dense model). To the best of our knowledge, only our manuscript and the EvoPress paper [4] are among the few works to apply structured pruning up to 70% sparsity. Higher sparsity levels are instead more common in unstructured pruning algorithms [2,3], where it is possible to better preserve task performance even at high sparsity (to note, however, that unstructured pruning allows limited inference speed-up and does not yield any GPU space storage reduction).

---

> > ### Author Response · Authors · 2025-06-17
> >
> > **References**
> >
> >
> > [1] Siddiqui, Shoaib Ahmed, et al. "A deeper look at depth pruning of LLMs." ICML 2024 Workshop on Theoretical Foundations of Foundation Models.\
> > [2] Frantar, Elias, and Dan Alistarh. "SparseGPT: Massive language models can be accurately pruned in one-shot." ICML 2023.\
> > [3] Sun, Mingjie, et al. "A Simple and Effective Pruning Approach for Large Language Models." ICLR 2024.\
> > [4] Sieberling, Oliver, et al. "EvoPress: Towards optimal dynamic model compression via evolutionary search." arXiv preprint arXiv:2410.14649 (2024).\
> > [5] Hoang, et al.  ”Revisiting pruning at initialization through the lens of Ramanujan graph." ICLR 2023.\
> > [6] Cunegatti, et al. "Understanding Sparse Neural Networks from their Topology via Multipartite Graph Representations." TMLR 2024.\
> > [7] Zhong, Longguang, et al. "BlockPruner: Fine-grained pruning for large language models." arXiv preprint arXiv:2406.10594 (2024).

---

### Review · Reviewer_QcEY · 2025-05-24

**Summary Of Contributions:**

This paper proposes a two-stage structured pruning framework (2SSP) combining width pruning (removing neurons from Feed-Forward Networks) and depth pruning (removing Attention submodules) for LLMs. The method balances pruning between these two stages using a novel sparsity allocation strategy based on perplexity. Experimental evaluations on four LLMs across various datasets and tasks demonstrate that 2SSP consistently outperforms several recent structured pruning methods, showing improved pruning runtime and model performance.

**Audience:**

Yes

**Broader Impact Concerns:**

No significant ethical concerns arise directly from the presented pruning method.

**Claims And Evidence:**

Yes

**Requested Changes:**

1. Clarify the motivation and theoretical reasoning behind selecting width and depth pruning specifically, rather than alternative structured pruning methods or combinations.

2. Expand comparisons to include recent state-of-the-art structured pruning methods [1-4] and newer LLM architectures like Llama 3 & 4 and Qwen 3.

[1] Dery, Lucio, et al. "Everybody prune now: Structured pruning of llms with only forward passes." arXiv preprint arXiv:2402.05406 (2024).

[2] Sengupta, Ayan, Siddhant Chaudhary, and Tanmoy Chakraborty. "You Only Prune Once: Designing Calibration-Free Model Compression With Policy Learning." The Thirteenth International Conference on Learning Representations 2025.

[3] Kong, Jun, et al. "Sample-aware Adaptive Structured Pruning for Large Language Models." Proceedings of the AAAI Conference on Artificial Intelligence. Vol. 39. No. 17. 2025.

[4] Le, Qi, et al. "Probe pruning: Accelerating LLMs through dynamic pruning via model-probing." The Thirteenth International Conference on Learning Representations 2025.

3. I suggest that the authors should also evaluate the method on "real" large LLMs (e.g., 60B–70B parameters) where model compression is more practically necessary. Additionally, assess higher sparsity levels (>50%) to determine if 2SSP remains effective under more extreme compression settings. Sparsity < 50% does not produce much practical usage IMO.

4. (minor) Consider conducting additional ablation studies with newer models to confirm robustness and effectiveness across contemporary architectures.

**Strengths And Weaknesses:**

Strengths:

S1. Seems the first work to combine width and depth pruning of LLMs.

S2. Extensive experimental setup, covering multiple LLM architectures, datasets, sparsity levels, and downstream tasks.

S3. Significant improvements demonstrated in both pruning runtime and model performance relative to the selected baseline methods.

Weaknesses:

W1. Technical novelty appears incremental, primarily combining existing methods without introducing fundamentally new insights or theoretical advancements.

W2. Motivation behind selecting specifically width and depth pruning methods is not clearly justified. Why not include other structured pruning methods or combinations?

W3. Empirical analysis could be more robust, notably the absence of comparisons with state-of-the-art pruning methods and recent and popular LLM architectures (e.g., Llama 3 & 4, Qwen 3, etc).

---

> ### Author Response · Authors · 2025-06-18
>
> We thank the reviewer for the valuable feedback. Below are our comments related to the Weaknesses and Requested Changes.
>
> **W1 - Novelty** While we are aware of different approaches that rely on magnitude for LLM pruning, as far as we know our approach is the first one to use activation magnitude to prune FeedForward submodules of Transformer blocks with the aim of preserving path information. To our knowledge, we are the first to apply magnitude, a well-established source of information for guiding network pruning, to Gated FeedForward submodules (i.e., the Gate, Up and Down projections, as described in Section 3) in order to remove in a structural manner rows and columns to maximize the network information flow. Furthermore, our proposed combination of depth and width pruning, including the empirical validation of the $\alpha$ parameter used to balance the contribution of each pruning scheme, is novel for what concerns one-shot training-free pruning algorithms for LLMs.
>
> **W2/C1 - Motivation** Concerning the motivation behind the choice of combining width and depth pruning, we kindly point the reviewer to our response **W2** to reviewer `Ngzt`.
>
> **W3/C2-C4 - Empirical analysis** We would like to point out that in Figure 3 (p. 7), the experimental results are available for a broader range of sparsity levels, up to 70%. The results depict the perplexity over WikiText-2 for 2SSP and the baselines.
> Also, we would like to thank the reviewer for pointing out that novel baselines and experiments over newer and larger models would improve our submission. However, while we were able to include results for the newer and larger models (see later), unfortunately we were not able to reproduce the suggested baselines, and we would like to clarify the main reason below. On the other hand, to address the reviewer’s comment, we still tried to make a direct comparison with those methods, based on the results available in their original papers.
>
> - [1] This paper is currently a pre-print, and the corresponding Github repository states “Bonsai [Still Under Construction]” (see https://github.com/ldery/Bonsai). Nevertheless, we include below a comparison of our perplexity results with the perplexity values (obtained on WikiText2 over Llama 1 and 2 at 50% sparsity) reported in [1], Figure 3 (to note that the results below are obtained with a sequence length of 1024, to align with the setup of [1]). While the numerical results are clearly in favor of Bonsai, we would like to highlight that, as specified by the authors in their paper, Bonsai requires approximately 4 hours to compress Llama2 7B. On the other hand, our approach runs on the same dataset, model, and sparsity in 1.5 minutes. Therefore, a direct comparison, taking into account both performance and pruning time, is, in our opinion, unfair (for the same reason, we selected as baselines only methods that require similar pruning times to our method).
>
> | Model | Method | WikiText2 | C4  | Runtime | GPUs |
> |---------------|--------|-------------|--------|-------|-------|
> | LLaMA-1 7B | Bonsai [1] | 15.72 | 22.31 | - | - |
> | | 2SSP | 34.75 | 39.25 | - | - |
> | LLaMA-2 7B | Bonsai [1] | 12.38 | 20.70 | $\sim$ 4hours |1xA6000 48G |
> | | 2SSP | 36.25 | 41.75 |2 minutes |1xA100 40GB |
>
> - [2] The publicly available repository only provides the code to prune the model (the README only provides the script to prune OPT-125M), while the evaluation details for loading and evaluating the sparse model are not provided (see the empty subsections **Evaluation script** and **Slicing the Attention** submodules at https://github.com/LCS2-IIITD/PruneNet). However, we compared our proposed method with the method from [2] by running the Zero-Shot experiments done in [2] under the same sparsity configuration. We report the results below. Out of 5 different Zero-Shot tasks, our method always outperforms the method from [2] in three tasks (PIQA, WinoGrande, and ARC-e), while over HellaSwag and ARC-c, our method is outperformed. It should be noted that PruneNet has been accepted at ICLR 2025 (conference 24th-28th April), while we submitted the manuscript to TMLR on the 5th of May.
>
>  Sparsity | Method | PIQA | WinoGrande | HellaSwag | ARC-e | ARC-c |
> |----------|-----------|-------|------------|-----------|-------|--------|
> | 20% | PruneNet [2] | 75.53 | 65.51 | **66.43** | 63.80 | **37.29**
> | | 2SSP | **75.68** | **65.58** | 50.20 | **69.31** | 35.15
> | 25% | PruneNet [2] | 72.09 | 62.43 | **62.33** | 60.14 | **36.18**
> | | 2SSP | **73.72** | **64.71** | 48.98 | **67.17** | 33.61 |
> | 30% | PruneNet [2] | 71.11 | 61.09 | **58.30** | 53.20 | **33.53**
> | | 2SSP | **72.68** | **63.14** | 45.57 | **60.31** | 28.24

---

> > ### Author Response · Authors · 2025-06-18
> >
> > - [3] The publicly available repository on GitHub https://github.com/JunKong5/AdaPruner is currently empty (despite the paper being accepted at AAAI 2025). We report below a comparison of Llama1 7B at 20% sparsity, taking the results from Table 1 in [3]. We can see that our proposed method (even if training-free) is able to outperform AdaPruner [3] in both the training-free and post-pruning fine-tuning cases.
> >
> > | Method | Post-Pruning Finetune | Perplexity on WikiText-2 |
> > |------|----|----|
> > | AdaPruner [3] | yes | 17.72 |
> > | AdaPruner [3] | no | 16.75 |
> > | 2SSP | no | **7.75** |
> >
> > - [4] is a dynamic pruning approach; hence, it prunes the model during inference. In other words, it selects the components to remove for every batch, rather than relying on a single calibration set to identify the critical components, hence trading the storage advantage of structured pruning (since every parameter needs to be stored in memory because it could be selected by a successive batch) for task performance improvement. Instead, 2SSP, as well as all the baselines tested in the paper, are static pruning approaches, allowing for memory reduction at the cost of slightly lower performance. For this reason, comparing methods belonging to the two categories would be somewhat unfair. Nevertheless, to address the reviewer’s suggestion, we include below a comparison of our results with the perplexity results (on WikiText-2) reported in [4], Tables 2 and 3, over Llama2 7B and Llama3 8B. As expected, [4] achieves better performance in terms of perplexity, but as said, comparing this method to static pruning approaches (such as our proposed 2SSP, which instead outperforms all other static pruning approaches, as shown in our paper) is unfair.
> >
> > **LLaMA-2 7B**
> > | Sparsity | Method |Perplexity on WikiText2|
> > |----------|--------|--------------|
> > | 20% | PP [4] | 8.10 |
> > | 20% | 2SSP | 8.50 |
> > |----------|--------|--------------|
> > | 40% | PP [4] | 16.80 |
> > | 40% | 2SSP | 17.50 |
> >
> > **LLaMA-3 8B**
> >
> > | Sparsity | Method |  Perplexity on WikiText2  |
> > |----------|--------|--------------|
> > | 20% | PP [4] | 9.30 |
> > | 20% | 2SSP |10.75 |
> > |----------|--------|--------------|
> > | 40% | PP [4] | 14.90 |
> > | 40% | 2SSP | 24.45 |
> >
> > To conclude, while we acknowledge the existence of these new methods, and we thank the reviewer once again for suggesting them, we would like to highlight that the baselines considered for the comparative analysis presented in our paper have been purposely selected because they have a comparable runtime w.r.t. our proposed method. In fact, all the depth-pruning techniques are fast pruning approaches (apart from EvoPress, which has been included due to the fact that its runtime could be reduced without significant performance degradation by changing the number of calibration data samples to give as input to the algorithm, see our response W1 to reviewer `Ngzt`), with SliceGPT being one of the fastest among the width-based approaches since it can obtain a sparse model over Llama 7b in $\sim$ 20 minutes over a single A100 40GB.
> > Concerning newer and larger models, as requested, we provide below the results on LLaMA3 8B, Qwen3 8B, and LLaMA3 70B (EvoPress and SliceGPT could not be run for the 70B models due to the large runtime and the unavailability of the SliceGPT code for LLaMA3). The numerical results reported below, in terms of perplexity, confirm the same trend shown for other models in the paper.
> >
> > **Llama3 8B**
> >
> > | Method | WikiText2 | C4 | Fineweb-Edu |
> > |--|-----------|--------|-------------|
> > | Dense| 6.31|8.22| 8.5|
> >
> > *Sparsity: 25%*
> >
> > | Method | WikiText2 | C4 | Fineweb-Edu |
> > |----------------|-----------|--------|-------------|
> > | ShortGPT | 3168.00 | 6304.00| 4080.00 |
> > | Sliding Window | 3168.00 | 6304.00| 4080.00 |
> > | BlockPruner | 13.81 | 20.38 | 17.13 |
> > | EvoPress | 13.56 | 19.50 | 15.19 |
> > | SliceGPT | 38.75 | 21.00 | 20.75 |
> > | **2SSP (Ours)**| **11.81** | **16.63** | **14.00** |
> >
> > *Sparsity: 37.5%*
> >
> > | Method | WikiText2 | C4 | Fineweb-Edu |
> > |----------------|-----------|--------|-------------|
> > | ShortGPT | 36352.00 | 9152.00| 8640.00 |
> > | Sliding Window | 36352.00 | 9152.00| 8640.00 |
> > | BlockPruner | 29.25 | 40.00 | 32.50 |
> > | EvoPress | 31.63 | 40.00 | 29.25 |
> > | SliceGPT | 84.50 | 37.00 | 36.25 |
> > | **2SSP (Ours)**| **21.38** | **24.25** | **22.00** |
> >
> > *Sparsity: 50%*
> >
> > | Method | WikiText2 | C4 | Fineweb-Edu |
> > |----------------|-----------|--------|-------------|
> > | ShortGPT | 2048.00 | 1864.00| 1752.00 |
> > | Sliding Window | 908.00 | 604.00 | 644.00 |
> > | BlockPruner | 158.00 | 123.00 | 115.50 |
> > | EvoPress | 108.50 | 108.50 | 96.00 |
> > | SliceGPT | 163.00 | 54.50 | 58.00 |
> > | **2SSP (Ours)**| **35.75** | **46.00** | **40.50** |

---

> > > ### Author Response · Authors · 2025-06-18
> > >
> > > **Qwen3 8B**
> > >
> > > Method | WikiText2 | C4 | Fineweb-Edu |
> > > |----------------|-----------|--------|-------------|
> > > | Dense| 10.25 | 14.67 | 11.06|
> > >
> > > *Sparsity: 25%*
> > >
> > > | Method | WikiText2 | C4 | Fineweb-Edu |
> > > |----------------|-----------|--------|-------------|
> > > | ShortGPT | 1097.00 | 518.00 | 320.50 |
> > > | Sliding Window | 1097.00 | 518.00 | 320.50 |
> > > | BlockPruner | 20.38 | 26.25 | 18.88 |
> > > | EvoPress | 19.75 | 27.00 | 19.13 |
> > > | SliceGPT | 46592.00 | 49664.00 | 38656.00 |
> > > | **2SSP (Ours)**| **14.94** | **21.38** | **16.63** |
> > >
> > > *Sparsity: 37.5%*
> > >
> > > | Method | WikiText2 | C4 | Fineweb-Edu |
> > > |----------------|-----------|--------|-------------|
> > > | ShortGPT | 4372.00 | 5152.00| 2652.00 |
> > > | Sliding Window | 514.00 | 227.25 | 202.88 |
> > > | BlockPruner | 54.50 | 54.50 | 40.50 |
> > > | EvoPress | 43.25 | 54.50 | 35.75 |
> > > | SliceGPT | 52736.00 | 46592.00 | 41216.00 |
> > > | **2SSP (Ours)**| **21.38** | **28.38** | **24.25** |
> > >
> > > *Sparsity: 50%*
> > >
> > > | Method | WikiText2 | C4 | Fineweb-Edu |
> > > |----------------|-----------|--------|-------------|
> > > | ShortGPT | 17296.00 | 12352.00| 11520.00 |
> > > | Sliding Window | 18256.00 | 5440.00 | 4404.00 |
> > > | BlockPruner | 191.00 | 174.00 | 139.00 |
> > > | EvoPress | 153.00 | 144.00 | 119.50 |
> > > | SliceGPT | 68096.00 | 56320.00 | 41216.00 |
> > > | **2SSP (Ours)**| **46.00** | **52.00** | **46.75** |

---

### Review · Reviewer_sCDe · 2025-06-09

**Summary Of Contributions:**

The paper investigates the joint application of width and depth pruning for structured compression of large language models (LLMs). While it claims to be the first to explore this combination, similar ideas have been studied in prior work under the frameworks of global and local sparsity allocation. The proposed method is empirically evaluated using both perplexity metrics and performance on downstream tasks.

**Audience:**

Yes

**Claims And Evidence:**

Yes

**Requested Changes:**

**1. Could the authors include comparisons with more state-of-the-art structured pruning methods?**
A more comprehensive evaluation against recent strong baselines—such as score-based, decomposition-based, or hypernetwork-based approaches—would strengthen the empirical validity of the paper.

**2. Could the authors provide more justification for the choice of specific width and depth pruning techniques, and their application to particular modules?**
It would be helpful to include theoretical motivations and/or thorough ablation studies to clarify why width pruning is applied to MLPs and depth pruning to attention layers.

**3. Could the authors investigate the scalability of the proposed method to larger-scale LLMs?**
Demonstrating the method’s effectiveness on models with tens of billions of parameters would provide a more comprehensive study on *Large* language models.

**Strengths And Weaknesses:**

**Strengths**:

- The paper considers both depth and width pruning together, applying them to different modules (attention and MLP) in Transformer-based LLMs. This joint perspective is relatively underexplored in the literature.
- The proposed pruning approach is evaluated using both perplexity and downstream task performance, providing empirical evidence for its effectiveness.

**Weaknesses**:
- Limited novelty beyond prior work: The technical contributions are incremental. Both width pruning and depth/layer pruning have been widely explored in prior works such as SliceGPT [1] and LLM-Pruner [2]. The paper does not introduce fundamentally new mechanisms or insights beyond these.
- Weak baseline comparisons: The paper lacks comparisons with several strong structured pruning baselines, such as: 1. OBD based methods: LLM-Surgeon [3] 2. Decomposition-based: MoDeGPT [4] 3. Layer pruning: SLEB [5] 4. Hypernetwork-based: DISP-LLM [6]
These methods represent their respective categories and have reported stronger performance on both perplexity and downstream tasks, making them valuablel baselines for a fair comparisons.
- Insufficient justification for module-specific pruning: The rationale behind applying width pruning only to MLP layers and depth pruning to attention layers is unclear. The paper does not provide sufficient theoretical motivation or ablation studies to support this design.
- No large-scale model validation: The experiments are limited to small and mid-sized LLMs. It remains unclear whether the proposed method can scale effectively to larger models (e.g., 30B+ parameters), which is important in applications.

References

[1] Saleh Ashkboos, Marco López-Croci, Matheus G. de Nascimento, Torsten Hoefler, James Hensman. SliceGPT: Compress Large Language Models by Deleting Rows and Columns. arXiv:2401.15024, 2024.

[2] Ma, Xinyin, Gongfan Fang, and Xinchao Wang. "Llm-pruner: On the structural pruning of large language models." Advances in neural information processing systems 36 (2023): 21702-21720.

[3] van der Ouderaa, Tycho FA, et al. "The llm surgeon." arXiv preprint arXiv:2312.17244 (2023).

[4] Lin, Chi-Heng, et al. "Modegpt: Modular decomposition for large language model compression." arXiv preprint arXiv:2408.09632 (2024).

[5] Song, Jiwon, et al. "Sleb: Streamlining llms through redundancy verification and elimination of transformer blocks." arXiv preprint arXiv:2402.09025 (2024).

[6] Gao, Shangqian, et al. "Disp-llm: Dimension-independent structural pruning for large language models." Advances in Neural Information Processing Systems 37 (2024): 72219-72244.

---

> ### Author Response · Authors · 2025-06-17
>
> We thank the reviewer for the valuable feedback. Below are our comments related to the Requested Changes.
>
> **C1 - Comparisons**: Below, we include the comparison over Llama2 7B with two of the requested baselines. Specifically, we include one width pruning approach (LLM-Pruner [2] as an example of OBD-based methods) and one depth pruning method (SLEB [5]). It can be seen that, for all three sparsity levels tested, our proposed method outperforms these two baselines.
> |Sparsity|Method|WikiText2|C4| Fineweb-Edu |
> |---|---|---|---|---|
> |0.25 |
> ||LLM-Pruner|10.54|12.32|10.19|
> ||SLEB|10.48|13.32|11.13|
> ||2SSP|**9.25**|**10.52**|**9.21**|
> |0.375|
> ||LLM Pruner|17.76|18.28|15.46|
> ||SLEB|22.76|24.18|19.81|
> ||2SSP|**14.64**|14.93|13.36|
> |0.5|
> ||LLM Pruner| 45.03 | 38.18 | 33.31 |
> ||SLEB| 105.62 | 77.31 | 67.43 |
> ||2SSP|**31.40**|**27.16**|**25.40**
>
> Unfortunately, we were not able to reproduce the other requested baselines, namely [3], [4], and [6]. Therefore, we compared the performance, pruning runtime, and GPUs usage based on the results taken from their corresponding original papers. The results in terms of perplexity on WikiText2 over Llama2 7B are reported in the table below.
>
> | Method | PPL. 20%  | PPL. 40% | PPL. 50%  | Runtime | GPUs |
> |---------------|--------|--------|--------|--------|--------|
> | LLM-Surgeon [3] | 8.18 | 10.39 | 15.38 | 17h08m | 4xH100 80 GB |
> | ModeGPT [4] | 6.16 | 8.41 | 11.88 | $\sim$ 5 hours | 4xA100 40GB |
> | Disp-LLM [6] | **6.10** | **8.11** | 9.80 | 8.83 hours| 2xH100 80 GB |
> | 2SSP | 8.50 | 17.50 | 31.40 | **[1.5,2.1] minutes** | **1xA100 40GB** |
>
> While the results in terms of perplexity are clearly not in favor of our method, we would like to highlight how our approach has a pruning runtime in the order of minutes, while the three above-mentioned baselines require hours, even running on much larger compute resources. For this reason, in our paper we purposely selected, among the baselines available in the recent literature, only those methods that achieve an effective tradeoff between pruning time and performance; namely, we only considered methods that, for a given model and sparsity rate, have a pruning time in the order of minutes. This is because our method is mainly built to achieve such a tradeoff, and comparing against similar tradeoff methods appears fairer than comparing against methods that have a much higher computational cost.
>
> **C2 - Justification**: The main justification for applying depth pruning only to Attention submodules is based on the observations made in [1], where it has been shown that depth pruning provides a lower performance gap when applied to the Attention submodules, while FeedForward submodules are more sensitive to depth removal. On the other hand, the reason why we applied width pruning only to FeedForward submodules is the fact that, while a path can be computed across FeedForward submodules (taking as neurons the activations of the hidden states), this is not possible for Attention submodules, because of the scaled dot-product between $Q$, $K$, and $V$ of the different heads. Preserving the path, however, is exactly the goal of our neuron pruning mechanism. For this reason, applying our proposed width pruning on Attention submodules is not possible. For further details about the motivation and the combination of depth and width pruning, we kindly point to our replies to reviewer `Ngzt` (W2).
> Finally, to expand our ablation studies, we performed an additional ablation study where we inverted the order of the two stages (first Attention pruning, then Neuron Pruning). We report below the numerical results, which show how this step inversion leads to worse results.
>
> |Sparsity|Method|WikiText|
> |---|---|---|
> 0.25
> || S2-S1 |10.75|
> ||S1–S2 (2SSP)|**9.25**|
> 0.375
> ||S2-S1|18.00|
> ||S1–S2 (2SSP)|**14.64**|
> 0.5
> ||S2-S1|47.50|
> ||S1–S2 (2SSP)|**31.40**|
>
> **C3 - Scalability**: We tested, as suggested, the scalability of 2SSP to larger models. In order to do that, we evaluated our proposed algorithm and the baselines over Llama3 70B. The results, reported below, indicate that the performance trend w.r.t. the baselines is in line with the results reported in the our paper (please note that EvoPress and SliceGPT could not be run for the 70B model due to, respectively, the large runtime and the unavailability of the SliceGPT code for Llama3). We would also like to highlight how, in such larger models, for a given sparsity rate, the drop of 2SSP w.r.t. the dense model is significantly smaller than that observed in the case of smaller models (such as the 7B models reported in the our paper).

---

> > ### Author Response · Authors · 2025-06-17
> >
> > #### Dense
> > | Model | WikiText2 | C4 | Fineweb-Edu |
> > |---------------------------|-----------|--------|-------------|
> > | meta-llama/Meta-Llama-3-70B | 2.98 | 7.38 | 5.66 |
> > ---
> > #### Sparsity: 25%
> > | Method | WikiText2 | C4 | Fineweb-Edu |
> > |----------------|-----------|--------|-------------|
> > | ShortGPT | 10.25 | 13.81 | 10.94 |
> > | Window-based | 18.25 | 20.13 | 15.63 |
> > | BlockPruner | 9.06 | 12.75 | 10.13 |
> > | **2SSP (Ours)**| **4.59** | **7.38** | **7.38** |
> > ---
> > #### Sparsity: 37.5%
> > | Method | WikiText2 | C4 | Fineweb-Edu |
> > |----------------|-----------|--------|-------------|
> > | ShortGPT | 304.00 | 90.00 | 58.00 |
> > | Window-based | 158.00 | 123.00 | 90.00 |
> > | BlockPruner | 12.94 | 17.50 | 13.81 |
> > | **2SSP (Ours)**| **6.47** | **9.63** | **9.31** |
> > ---
> > #### Sparsity: 50%
> > | Method | WikiText2 | C4 | Fineweb-Edu |
> > |----------------|-----------|--------|-------------|
> > | ShortGPT | 1240.00 | 776.00 | 604.00 |
> > | Window-based | 2400.00 | 1696.00| 752.00 |
> > | BlockPruner | 28.75 | 32.50 | 24.25 |
> > | **2SSP (Ours)**| **11.25** | **14.69** | **14.25** |
> >
> > **References**
> >
> > [1] Saleh Ashkboos, Marco López-Croci, Matheus G. de Nascimento, Torsten Hoefler, James Hensman. SliceGPT: Compress Large Language Models by Deleting Rows and Columns. arXiv:2401.15024, 2024.\
> > [2] Ma, Xinyin, Gongfan Fang, and Xinchao Wang. "LLM-pruner: On the structural pruning of large language models." Advances in neural information processing systems 36 (2023): 21702-21720.\
> > [3] van der Ouderaa, Tycho FA, et al. "The LLM Surgeon." The Twelfth International Conference on Learning Representations.\
> > [4] Lin, Chi-Heng, et al. "ModeGPT: Modular decomposition for large language model compression." arXiv preprint arXiv:2408.09632 (2024).\
> > [5] Song, Jiwon, et al. "SLEB: Streamlining LLMs through Redundancy Verification and Elimination of Transformer Blocks." arXiv preprint arXiv:2402.09025 (2024)\
> > [6] Gao, Shangqian, et al. "Disp-LLM: Dimension-independent structural pruning for large language models." Advances in Neural Information Processing Systems 37 (2024): 72219-72244.

---

### Author Response · Authors · 2025-06-18

Dear Action Editor and Reviewers,

We have just posted our point-to-point replies to the comments and suggestions indicated in each of the reviews. We did our best to address all of them. We look forward to receiving further feedback, and we are open to providing any other information if needed.

Kind regards,

The authors

---

### Decision · Action_Editor_kXbM · 2025-08-07

**Recommendation:** Accept with minor revision

**Additional Comments:**

This paper presents a solid and practical contribution with a well-executed empirical study, but the technical novelty is incremental—combining two well-known pruning strategies in a careful, though not fundamentally new, way. While the proposed balancing scheme and the design rationale are reasonable, the lack of deeper theoretical analysis or novel pruning algorithms somewhat limits the innovation.

Additionally, although the authors have gone to great lengths to extend their comparisons and ablation studies in the rebuttal, the original submission would benefit from clearer upfront justifications for pruning module choices and broader sparsity exploration. These are relatively minor revisions and do not detract from the overall value of the work.

**Audience:**

Yes

**Audience Explanation:**

Structured pruning of LLMs remains a highly active area of research, especially in contexts where resource-constrained deployment is crucial. TMLR's readership, which includes researchers and practitioners working on model compression, deployment, and LLM efficiency, will likely find this work of interest.

The combination of fast, training-free compression with a performance-competitive strategy is especially relevant for practitioners who require efficient, interpretable, and adaptable pruning methods. Furthermore, the generality of the method across different LLM architectures and the empirical focus on real-world usability make this paper attractive beyond a narrow theoretical audience.

**Claims And Evidence:**

Yes

**Claims Explanation:**

The authors present thorough empirical evidence supporting their key claims. They evaluate 2SSP across four different LLM families (including recent models like LLaMA3 and Qwen3), multiple sparsity levels (up to 70%), and both language modeling and downstream tasks. The results demonstrate that 2SSP consistently outperforms several fast, structured pruning baselines in terms of perplexity and task accuracy, while also drastically reducing pruning runtime.

The authors are transparent about trade-offs — such as minor performance losses compared to slower, compute-intensive methods—and clearly articulate their design choices. Additionally, they provide extensive ablation studies, scalability tests, and baseline comparisons (including difficult-to-reproduce methods using reported numbers), which further reinforce the validity and rigor of their empirical evaluation.